# A frameshift in *Yersinia pestis rcsD* alters canonical Rcs signalling to preserve flea-mammal plague transmission cycles

Xiao-Peng Guo[1†], Hai-Qin Yan[2,3†], Wenhui Yang[4†], Zhe Yin[4], Viveka Vadyvaloo[3*], Dongsheng Zhou[4*], Yi-Cheng Sun[1*]

[1]NHC key laboratory of Systems Biology of Pathogens, Institute of Pathogen Biology, Chinese Academy of Medical Sciences and Peking Union Medical College, Beijing, China; [2]Department of Basic Medical Sciences, Anhui Key Laboratory of Infection and Immunity, Bengbu Medical College, Bengbu, China; [3]Paul G. Allen School for Global Health, Washington State University, Pullman, United States; [4]State Key Laboratory of Pathogen and Biosecurity, Beijing Institute of Microbiology and Epidemiology, Beijing, China

*For correspondence:
vvadyvaloo@wsu.edu (VV);
zhouds@bmi.ac.cn (DZ);
sunyc@ipbcams.ac.cn (YCS)

†These authors contributed equally to this work

Competing interest: The authors declare that no competing interests exist.

**Abstract** Multiple genetic changes in the enteric pathogen *Yersinia pseudotuberculosis* have driven the emergence of *Yesinia pestis*, the arthropod-borne, etiological agent of plague. These include developing the capacity for biofilm-dependent blockage of the flea foregut to enable transmission by flea bite. Previously, we showed that pseudogenization of *rcsA*, encoding a component of the Rcs signalling pathway, is an important evolutionary step facilitating *Y. pestis* flea-borne transmission. Additionally, *rcsD*, another important gene in the Rcs system, harbours a frameshift mutation. Here, we demonstrated that this *rcsD* mutation resulted in production of a small protein composing the C-terminal RcsD histidine-phosphotransferase domain (designated RcsD-Hpt) and full-length RcsD. Genetic analysis revealed that the *rcsD* frameshift mutation followed the emergence of *rcsA* pseudogenization. It further altered the canonical Rcs phosphorylation signal cascade, fine-tuning biofilm production to be conducive with retention of the *pgm* locus in modern lineages of *Y. pestis*. Taken together, our findings suggest that a frameshift mutation in *rcsD* is an important evolutionary step that fine-tuned biofilm production to ensure perpetuation of flea-mammal plague transmission cycles.

## Editor's evaluation

This is a valuable study that provides convincing evidence for fine-tuning of a signal transduction pathway in the emergence of the bacterial pathogen *Yersinia pestis*. The work advances our understanding of bacterial signal transduction, and will be of interest to those studying the evolution of *Y. pestis* and its adaptation to flea-borne transmission.

## Introduction

Approximately 6000–7000 years ago, *Yersinia pestis* evolved to be an arthropod-borne pathogen from its ancestor *Yersinia pseudotuberculosis* (**Cui et al., 2013**; **Rasmussen et al., 2015**; **Spyrou et al., 2018**). Despite their recent divergence, these species have markedly different life cycles. *Y. pseudotuberculosis* is transmitted by the faecal-oral route and usually causes a mild, self-limiting enteric disease in mammals (**Putzker et al., 2001**). *Y. pestis*, uniquely amongst enteric Gram-negative bacteria, is highly virulent and relies on flea-borne transmission (**Hinnebusch, 2005**). The co-occurrence of both

**eLife digest** *Yersinia pestis*, the agent responsible for the plague, emerged 6,000 to 7,000 years ago from *Yersinia pseudotuberculosis*, another type of bacteria which still exists today. Although they are highly similar genetically, these two species are strikingly different. While *Y. pseudotuberculosis* spreads via food and water and causes mild stomach distress, *Y. pestis* uses fleas to infect new hosts and has killed millions.

A small set of genetic changes has contributed to the emergence of *Y. pestis* by allowing it to thrive inside a flea and maximise its transmission. In particular, some of these mutations have led to the bacteria being able to come together to form a sticky layer that adheres to the gut of the insect, with this 'biofilm' stopping the flea from feeding on blood. The starving flea keeps trying to feed, and with each bite comes another opportunity for *Y. pestis* to jump host. However, it remains unclear exactly how the mutations have influenced biofilm formation to allow for this new transmission mechanism to take place.

To examine this phenomenon, Guo et al. focused on *rcsD*, a gene that codes for a component of the signalling system that controls biofilm creation. In *Y. pestis* this sequence has been mutated to become a 'pseudogene', a type of sequence which is often thought to be non-functional. However, the experiments showed that, in *Y. pestis*, *rcsD* could produce small amounts of a full-length RcsD protein similar to the one found in *Y. pseudotuberculosis*. However, the gene mostly produces a short 'RcsD-Hpt' protein that can, in turn, alter the expression of many genes, including those that decrease biofilm formation. This may prove to be beneficial for *Y. pestis*, for example when the bacteria switches from living in fleas to living in humans, where it does not require a biofilm.

Guo et al. further investigated the impact of *rcsD* becoming a pseudogene in *Y. pestis*, showing that if normal amounts of the full-length RcsD protein are produced, the bacteria quickly lose the gene that allows them to form biofilm in fleas, and cause disease in humans. In fact, additional analyses revealed that all sequenced strains of ancient and modern *Y. pestis* bacteria can produce RcsD-Hpt, even if they do not carry the same exact *rcsD* mutation. Overall, these results indicate that *rcsD* turning into a pseudogene marked an important step in the emergence of *Y. pestis* strains that can cause lasting plague outbreaks. They also point towards pseudogenes having more important roles in evolution than previously thought.

ancestor and descendant *Yersinia* species provides an exemplary model to study the evolution of bacterial pathogens (*Hinchliffe et al., 2003*; *Hinnebusch, 1997*; *Wren, 2003*).

*Y. pestis* is transmitted by flea bites via a crude regurgitation mechanism (*Hinnebusch and Erickson, 2008*). After entering the flea gut, the planktonic *Y. pestis* form large aggregates and colonize the proventriculus. When infected fleas feed again as early as 1–3 days post infection, the bacterial mass in the proventriculus is regurgitated into the bite site, leading to a phenomenon referred to as early phase transmission (*Bosio et al., 2020*; *Dewitte et al., 2020*; *Eisen et al., 2007*; *Hinnebusch and Erickson, 2008*). Concurrently, *Y. pestis* continues to multiply in the flea digestive tract and forms HmsHFRS-dependent biofilms in the proventriculus, which blocks flea feeding (*Abu Khweek et al., 2010*). Continuous attempts to feed by the starved flea promotes reflux of bacteria-contaminated blood to the bite site, a phenomenon termed biofilm-dependent transmission (*Bland et al., 2018*). In contrast to the low efficiency of early phase transmission, which usually requires several infected fleas simultaneously feeding on a naïve host, biofilm-dependent late-stage transmission is highly efficient, and a single blocked flea has high potential for transmission (*Bosio et al., 2020*).

*Y. pestis* diverged from *Y. pseudotuberculosis* through a series of gene gains, gene losses, and genome rearrangements (*Cao et al., 2022*; *Hinchliffe et al., 2003*; *McNally et al., 2016*; *Rascovan et al., 2019*; *Sun et al., 2014*). Acquisition of the *ymt* gene enabled *Y. pestis* to survive and colonize the flea midgut to sustain flea-borne plague through expansion of its mammalian host range (*Bland et al., 2021*; *Sun et al., 2014*). Loss of the three genes, *rcsA*, *pde2*, and *pde3*, altered the c-di-GMP signalling pathway, which increased biofilm-forming capability (*Sun et al., 2008*; *Sun et al., 2014*). These four genetic changes enabled the *Y. pseudotuberculosis* progenitor strain to form biofilms in the proventriculus of fleas, promoting a flea-borne transmission modality (*Sun et al., 2014*).

The Rcs phosphorelay system, a non-orthodox two-component signal transduction system, consists of a hybrid sensor kinase RcsC, the phosphotransfer protein RcsD, and a response regulator RcsB (*Guo and Sun, 2017*; *Wall et al., 2018*). In Enterobacteriaceae, the outer membrane (OM) lipoprotein, RcsF, senses OM- and peptidoglycan-related stress (*Smith et al., 2021*; *Tata et al., 2021*). The inner membrane protein, IgaA, subsequently relays these signals to RcsD, which activates the Rcs phosphorelay system (*Cho et al., 2014*; *Wall et al., 2020*). Autophosphorylated RcsC transfers a phosphate group to a conserved histidine residue in the C-terminal histidine-phosphotransferase (HPt) domain of RcsD, which is finally transferred to RcsB (*Takeda et al., 2001*). Phosphorylated RcsB acts either alone or in combination with auxiliary proteins to regulate expression of target genes (*Clarke, 2010*; *Huesa et al., 2021*; *Pannen et al., 2016*). Functional RcsA works in concert with RcsB as a heterodimer to inhibit *Y. pseudotuberculosis* biofilm formation, in part by repressing expression of the c-di-GMP synthesis genes *hmsT* and *hmsD* (*Bobrov et al., 2011*; *Fang et al., 2015*; *Guo et al., 2015*; *Sun et al., 2012*; *Sun et al., 2011*). In *Y. pestis*, however, RcsA (RcsA_pe) was disrupted by acquiring a 30 bp repeat insertion sequence, leading to enhanced capacity for biofilm formation (*Sun et al., 2008*). *rcsD* in *Y. pestis* (*rcsD_pe*) is a putative pseudogene due to a 1 bp deletion, but it retains a limited ability to modulate biofilm formation *in vitro* (*Sun et al., 2008*).

Here, we investigated the functional consequences of the frameshift mutation in *rcsD_pe* and the mechanism by which it modulates activity of the Rcs signalling system. We found that the *rcsD_pe* variant produces a 103-amino acid protein containing the C-terminal HPt domain of RcsD, designated as RcsD-Hpt, and that this protein plays a dominant role in Rcs signalling in *Y. pestis*. Frameshifted *rcsD* alters Rcs signal transduction, subsequently modulating the expression of dozens of genes and the capacity for biofilm formation in *Y. pestis*. These evolutionary events may represent an important step in the emergence of ubiquitous branches of *Y. pestis* that can capably maintain plague outbreaks.

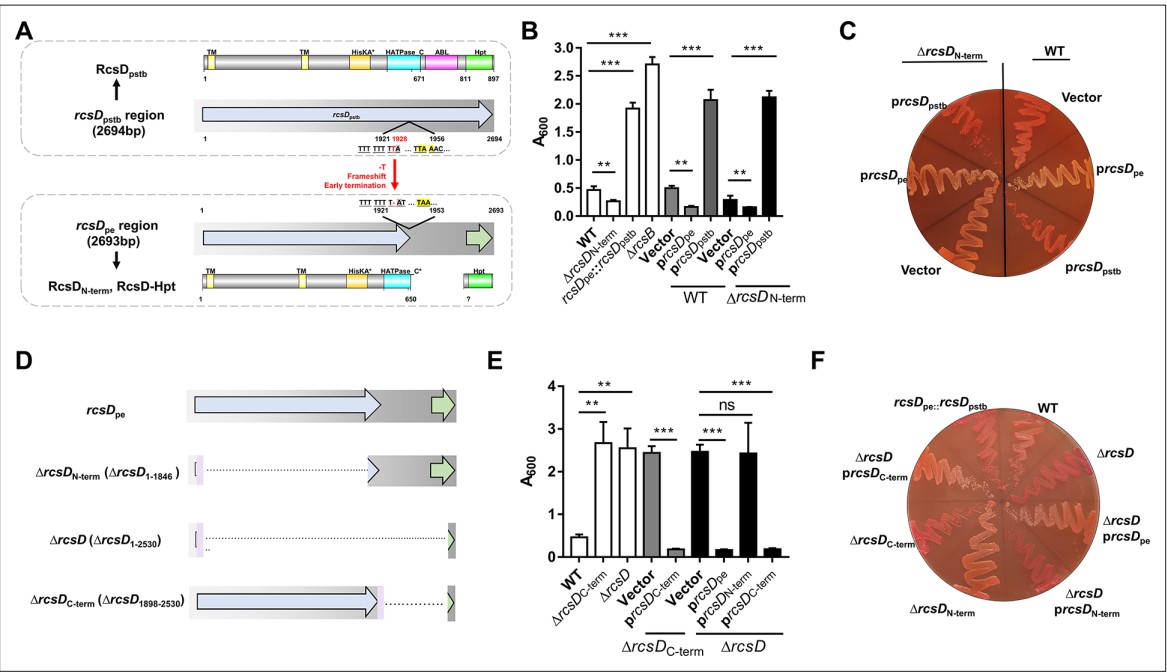

**Figure 1.** *rcsD_pe* negatively regulates biofilm formation, while *rcsD_pstb* positively regulates biofilm formation in *Y. pestis*. (**A**) Schematic representation of the *rcsD* frameshift mutation that occurred during speciation of *Y. pestis* from its ancestor *Y. pseudotuberculosis*. *rcsD_pstb*: *rcsD* from *Y. pseudotuberculosis*; *rcsD_pe*: *rcsD* from *Y. pestis*; △*rcsB*, *rcsB* deletion; RcsD_N-term: N-terminal fragment of RcsD; RcsD-Hpt: C-terminal HPt domain of RcsD. Crystal violet (CV) binding assay (**B**) and Congo red (CR) pigmentation assay (**C**) using *Y. pestis* KIM6+ (WT), an N-terminal deletion mutant (△*rcsD_N-term*), and an *rcsD_pstb* substitution strain (*rcsD_pe::rcsD_pstb*) and their derivatives that carry plasmids harbouring *rcsD_pe* (p*rcsD_pe*) or *rcsD_pstb* (p*rcsD_pstb*). (**D**) Schematic representation of the *rcsD* mutations constructed in this study. CV binding assay (**E**) and CR pigmentation assay (**F**) using *Y. pestis* KIM6+ (WT) and its derivative strains harbouring plasmids expressing different truncations of RcsD. CV assays in panels B and E were performed together. Error bars represent ± SD from three independent experiments with three replicates. Statistical analysis was performed using one-way analysis of variance (ANOVA) with Dunnett's multiple comparisons post-test. ns, not significant; *p<0.05, **p<0.01, ***p<0.001.

# Results

## $rcsD_{pe}$ negatively regulates, while $rcsD_{pstb}$ positively regulates, biofilm formation in *Y. pestis*

RcsD, an inner membrane protein, has an HPt domain at its C-terminus (*Takeda et al., 2001*; *Wall et al., 2018*). The *rcsD* gene in *Y. pestis* has undergone a frameshift after codon 642 due to a single nucleotide deletion (*Figure 1A*). However, $rcsD_{pe}$ is still functional as deletion of the N-terminal region (1–1846 bp, $\Delta rcsD_{N\text{-term}}$) of $rcsD_{pe}$ reduced biofilm formation and Congo red (CR) pigmentation (*Figure 1B and C*; *Sun et al., 2008*), while replacement of $rcsD_{pe}$ with $rcsD_{pstb}$, and *rcsB* deletion significantly increased biofilm formation (*Figure 1B*; *Sun et al., 2008*). To further characterize the differences between $rcsD_{pe}$ and $rcsD_{pstb}$, biofilm formation was determined in *Y. pestis* wild type and *rcsD* N-terminal deletion strains carrying plasmid-encoded copies of each gene, respectively (*Figure 1B and C*). Consistent with our previous report (*Sun et al., 2008*), overexpression of $rcsD_{pstb}$ significantly increased CR pigmentation and biofilm formation *in vitro* (*Figure 1B and C*). By contrast, overexpression of $rcsD_{pe}$ decreased the CR phenotype and formation of biofilm (*Figure 1B and C*), indicating that $rcsD_{pe}$ might play a different role to $rcsD_{pstb}$ in the modulation of Rcs signalling.

Next, we constructed two mutants, one lacking the entire $rcsD_{pe}$ gene, and the other lacking a C-terminal fragment ($\Delta rcsD$ and $rcsD_{C\text{-term}}$) (*Figure 1D*). In contrast to the *rcsD* N-terminal deletion mutant ($\Delta rcsD_{N\text{-term}}$), these two strains showed comparable phenotypes with increased CR adsorption and *in vitro* biofilm formation (*Figure 1E and F*). The opposing effects on biofilm formation in these mutants suggest that the C-terminal region may contribute to the inhibition of biofilm formation. To test this, we expressed $rcsD_{C\text{-term}}$, $rcsD_{N\text{-term}}$, and $rcsD_{pe}$ in a *rcsD* deletion mutant. Overexpression of $rcsD_{C\text{-term}}$ and $rcsD_{pe}$ but not $rcsD_{N\text{-term}}$ greatly decreased CR pigmentation and biofilm formation (*Figure 1E and F*). Collectively, these results demonstrate that RcsD$_{pstb}$ positively regulates, while the frameshifted $rcsD_{pe}$ negatively regulates, biofilm formation, and that this activity depends on the C-terminus.

## $rcsD_{pe}$ expresses intact RcsD and a small HPt-containing domain protein RcsD-Hpt

The C-terminal HPt domain of *rcsD* is phosphorylated by RcsC and subsequently transfers a phosphate group to the response regulator RcsB *in vitro* (*Takeda et al., 2001*). Given that frameshifted $rcsD_{pe}$ is functional, and expression of the C-terminus of *rcsD* is sufficient to repress biofilm formation, we hypothesized that a small protein containing the HPt domain is expressed by $rcsD_{pe}$. To identify the putative open reading frame (ORF) of this latter protein, we analysed candidate translational initiation sites in the C-terminus of $rcsD_{pe}$ by searching for AUG, GUG, and UUG codons, which account for approximately 80%, 12%, and 8% of start codons in bacterial genomes (*Srivastava et al., 2016*). We found two putative ORFs (encoding genes composing 462 and 573 nucleotides, respectively, *Figure 2—figure supplement 1A*) with UUG start codons, denoted as UUG$^{-462}$ and UUG$^{-573}$. Ectopic expression of these two putative ORFs in *Y. pestis* KIM6+ showed comparable biofilm formation to expression of full-length $rcsD_{pe}$ (*Figure 2—figure supplement 1B*). Mutation of these two start codons alone (UUG$^{-462}$→UUA, UUG$^{-573}$→CUU) or together did not alter the function of $rcsD_{pe}$ (*Figure 2—figure supplement 1C*), indicating they are not the start sites promoting expression of the functional protein.

AUU, CUG, and AUC are occasionally used as start codons in bacteria (*Cao and Slavoff, 2020*; *D'Lima et al., 2017*). We identified a putative ORF, encoding a 103-residue protein, initiated by an AUU codon, which also had a predicted RBS binding site (*Figure 2A*). Mutation of this codon or the predicted RBS region in $rcsD_{pe}$ abolished its function, while replacement of AUU with a strong AUG start codon significantly enhanced its function (*Figure 2B and C*). Furthermore, ectopic expression of the 103-residue HPt domain with an AUU start codon, but not with a GGU codon, using a modified pBAD vector in the *Y. pestis* KIM6+ wild type strain, or *rcsD* deletion mutant, strongly repressed biofilm formation and CR pigmentation (*Figure 2—figure supplement 1D*), indicating that AUU is a functional start codon initiating translation of RcsD-Hpt.

Next, we introduced 3xFlag and 6xHis tags to $rcsD_{pe}$ and $rcsD_{pstb}$ and analysed their expression by western blotting. A band in the tagged RcsD$_{pstb}$ strain was observed at the expected size (*Figure 2D, Lane3*), but no visible band was detected in the tagged $rcsD_{pe}$ strain (*Figure 2D, Lane2*). To enhance detection sensitivity, the $rcsD_{pe}$ lysates were enriched before western blotting. An approximately

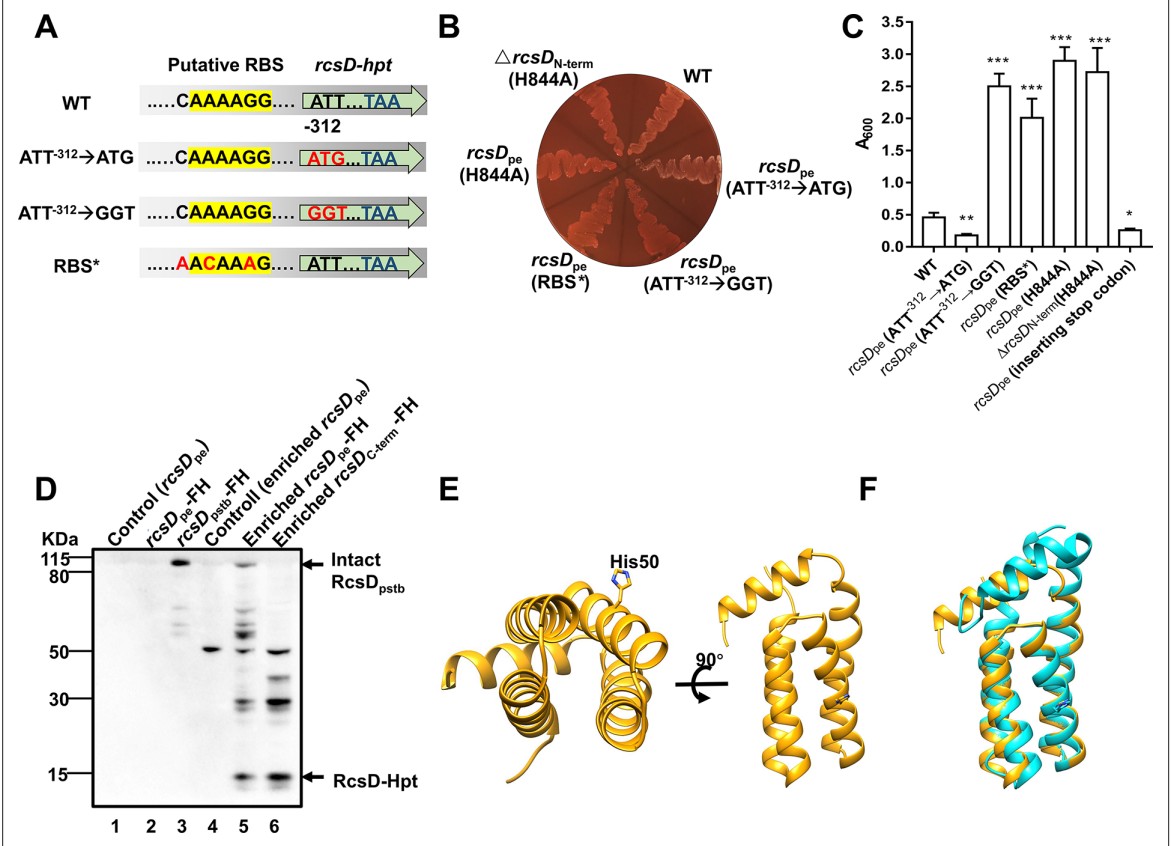

**Figure 2.** Expression of RcsD-Hpt is initiated by an uncommon AUU start codon and negatively regulates biofilm formation in *Y. pestis*. (**A**) Schematic representation of mutations introduced to the putative RBS and start codon of RcsD-Hpt. ATT$^{-312}$→GGT or ATT$^{-312}$→ATG indicate that the predicted ATT start codon was mutated to GGT or ATG, respectively. RBS* indicates that the putative RBS of RcsD-Hpt was mutated. Congo red (CR) pigmentation assay (**B**) and crystal violet (CV) binding assay (**C**) using *Y. pestis* KIM6+ (WT) and its derivative strains. Error bars represent ± SD from three independent experiments with three replicates. Statistical analysis was performed using one-way analysis of variance (ANOVA) with Dunnett's multiple comparisons post-test, comparing each construct with WT. *p<0.05, **p<0.01, ***p<0.001. (**D**) Expression of RcsD in *Y. pestis* and its derivatives were detected by western blot analysis using an anti-Flag antibody (see *Figure 2—source data 1* details). The 3xFlag and His6 epitope tags (FH) encoding sequences were fused to the C-terminus of *rcsD*$_{pe}$ and *rcsD*$_{pstb}$. (**E**) Structure of RcsD-Hpt (103 residues) predicted by AlphaFold2. Conserved His50 is located at the α3 helix of RcsD-Hpt. (**F**) Structure comparison of RcsD-Hpt (103 residues) by AlphaFold2 (yellow) and HptB of *Pseudomonas aeruginosa* PAO1 (PDB 7C1I, cyan).

The online version of this article includes the following source data and figure supplement(s) for figure 2:

**Source data 1.** Raw source data for *Figure 2D*.

**Figure supplement 1.** Characterization of proteins encoded by *rcsD*$_{pe}$ region, related to *Figure 2*.

**Figure supplement 1—source data 1.** Raw source data for *Figure 2—figure supplement 1E*.

**Figure supplement 1—source data 2.** Raw source data for *Figure 2—figure supplement 1F*.

**Figure supplement 2.** The conserved His residue, H844, is important for the function of RcsD and RcsD-Hpt.

15 kDa band, corresponding to the predicted protein containing the HPt domain, was detected in both *rcsD*$_{pe}$, *rcsD*$_{C-term}$, and *rcsD*$_{pstb}$ expressing strains (*Figure 2D* and *Figure 2—figure supplement 1E*, see *Figure 2—source data 1*, *Figure 2—figure supplement 1—source data 1* for details), but not detected in the putative start codon mutated *rcsD*$_{pe}$ expressing strain (*Figure 2—figure supplement 1E*, see *Figure 2—figure supplement 1—source data 1* for details). In addition, full-length RcsD was detected in *rcsD*$_{pe}$ but not in *rcsD*$_{C-term}$ and *rcsD*$_{pe}$-stop (a stop codon was introduced upstream of the frameshift site) expressing strains (*Figure 2D* and *Figure 2—figure supplement 1F*, see *Figure 2—source data 1*, *Figure 2—figure supplement 1—source data 2* for details), indicating that intact RcsD is weakly expressed in wild type *Y. pestis* through translational recoding (*Rodnina et al., 2020*). This is in accordance with the translational *lacZ* reporter system, which detected 1% readthrough once

frameshifted (*Figure 2—figure supplement 1G*). Consistent with these results, *Y. pestis* with *rcsD*pe-stop showed similar biofilm and CR phenotype as the *rcsD*N-term deletion strain (*Figure 2C* and data not shown). Taken together, these data suggest that a 103-amino acid protein, designated as RcsD-Hpt, is expressed by *rcsD*pe and functions as a negative regulator of biofilm formation in *Y. pestis*.

RcsD is a phosphorelay protein that can transfer phosphate from the conserved His residue in its HPt domain to a conserved Asp in the receiver domain of RcsB and can also dephosphorylate this site (*Ancona et al., 2015*; *Takeda et al., 2001*). An H844A mutation in the *Y. pestis* wild type or *rcsD*N-term deletion strain showed a comparable phenotype to a full *rcsD* deletion strain (*Figure 2B and C* and *Figure 1E*), while plasmid-borne expression of the mutated version of *rcsD*pe or *rcsD-hpt* in *Y. pestis* wild type and its *rcsD* mutants displayed a similar phenotype to an empty vector control (*Figure 2—figure supplement 2A*). In addition, plasmid-borne expression of the mutated version of *rcsD*pstb in the *Y. pestis* *rcsD*N-term deletion strain showed significantly less biofilm than that of *rcsD*-pstb (*Figure 2—figure supplement 2B*), indicating H844 might be important for the dephosphorylation ability of RcsD. Next, we modelled the structure of RcsD-Hpt with AlphaFold2 (*Fowler and Williamson, 2022*; *Figure 2E*), which revealed a similar structure to HptB, an HPt orphan protein in *P. aeruginosa* (*Figure 2F*; *Chen et al., 2020*). Like HptB, the predicted structure of RcsD-Hpt shows an elongated bundle of four helices α2, α3, α4, and α5, covered by the short N-terminal α1 helix. The imidazole side chain of the conserved active-site histidine residue His50 (His844 in RcsD*pstb*) is located near the middle of helix α3 and protrudes from the bundle where it is exposed, as His57 is in HptB. Taken together, RcsD-Hpt may function as a classical HPt orphan protein, and the conserved His residue is crucial for its function.

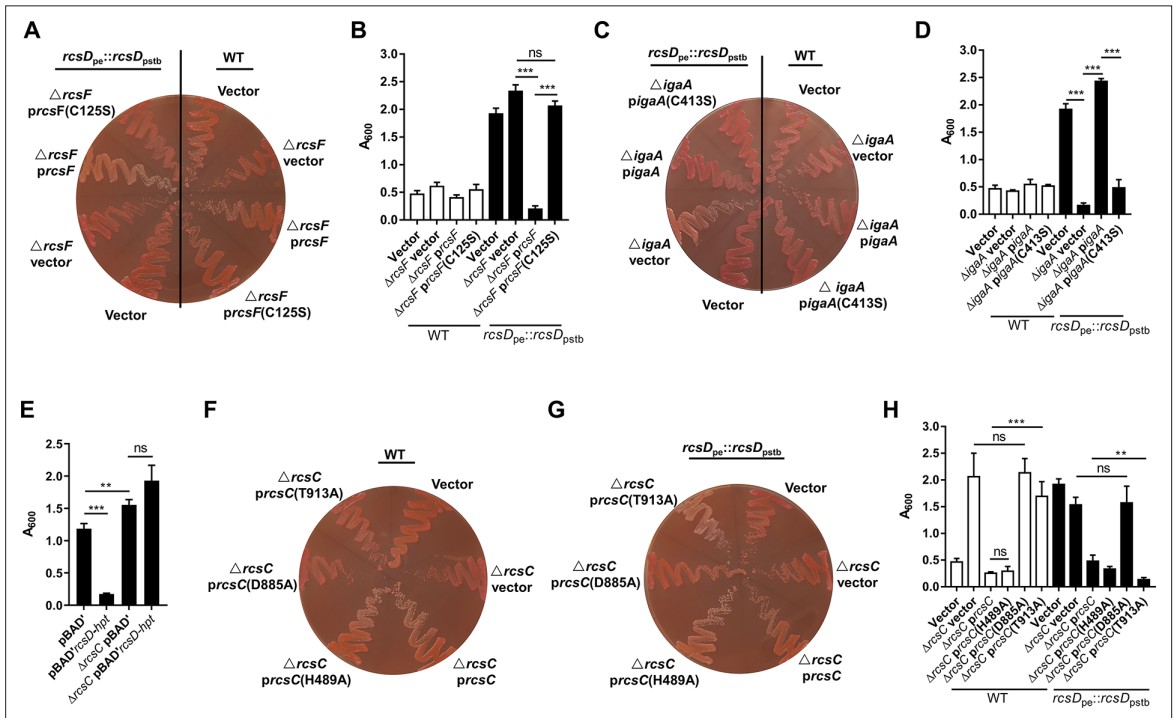

**Figure 3.** The frameshift mutation in *rcsD* alters the Rcs signalling pathway in *Y. pestis*. Congo red (CR) pigmentation assay (**A and C**) and crystal violet (CV) biofilm assay (**B and D**) the derivatives of *Y. pestis* KIM6+ (wild type [WT]) and the *rcsD*pstb substitution strain. p*rcsF*, plasmid expressing *rcsF*; p*rcsF* (C125S), plasmid expressing *rcsF* with cysteine (**C**) to serine (**S**) substitution at position 125; p*igaA*, plasmid expressing *igaA*; p*igaA* (C413S), plasmid expressing *igaA* with cysteine (**C**) to serine (**S**) substitution at position 413. (**E**) CV binding assay using WT and a *rcsC* deletion mutant expressing RcsD-Hpt (*rcsD-hpt*). CR pigmentation assay (**F and G**) and CV biofilm assay (**H**) using two *Y. pestis* *rcsC* deletion mutants expressing different *rcsC* variants. CV assays in panels B, D, and H were performed together. Error bars represent ± SD from three independent experiments with three replicates. Statistical analysis was performed using one-way analysis of variance (ANOVA) with Dunnett's multiple comparisons post-test. ns, not significant; *p<0.05, **p<0.01, ***p<0.001.

## A frameshift in *rcsD* alters Rcs signalling in *Y. pestis*

RcsF and IgaA, which regulate environmental stress sensing (*Cho et al., 2014*; *Guo and Sun, 2017*; *Wall et al., 2018*), transfer signals to RcsD through interaction of IgaA with its periplasmic domain (*Wall et al., 2020*). Given that RcsD-Hpt does not encode a functional periplasmic domain, we hypothesized that the roles of RcsF and IgaA are dispensable in the *Y. pestis* Rcs signalling system. We constructed *rcsF* and *igaA* deletion mutants in the *Y. pestis* wild type and $rcsD_{pstb}$ substitution strains, respectively. In the *Y. pestis* $rcsD_{pstb}$ substitution strain, deletion of *rcsF* increased CR adsorption and biofilm formation (*Figure 3A and B*), while deletion of *igaA* completely abolished biofilm formation and CR binding (*Figure 3C and D*). Furthermore, overexpression of RcsF but not its C125S mutant (as *Escherichia coli* RcsF cysteine mutants are inactive *Rogov et al., 2011*), decreased biofilm formation and CR adsorption (*Figure 3A and B*), while expression of IgaA but not its C413S mutant (the conserved cysteine is essential for the function of IgaA in *Salmonella enterica Pucciarelli et al., 2017*) complemented the phenotype of the *igaA* deletion strain (*Figure 3C and D*). These results indicate that when receiving signals transduced from RcsF and IgaA, $RcsD_{pstb}$ mediates decreased phosphorylation of RcsB, and thus Rcs signalling is switched off. Alternately, deletion or ectopic expression of *rcsF* or *igaA* does not regulate biofilm formation and CR pigmentation in wild type *Y. pestis* KIM6+ (*Figure 3A–D*), suggesting a decoupling of the requirement for RcsF and IgaA and the Rcs system after the transition from $rcsD_{pstb}$ to $rcsD_{pe}$.

RcsC, a bifunctional histidine kinase and phosphatase, phosphorylates and dephosphorylates RcsD, which subsequently, via RcsB, activates or represses expression of its target genes (*Guo and Sun, 2017*; *Wall et al., 2018*). Deletion of RcsC resulted in increased biofilm formation and CR pigmentation (*Sun et al., 2008*; *Figure 3E*), indicating that RcsC is involved in Rcs signalling in *Y. pestis*. In addition, high expression of RcsD-Hpt in the *rcsC* deletion mutant did not affect biofilm formation and pigmentation (*Figure 3E*), indicating that RcsC is required for RcsD-Hpt-mediated biofilm repression in *Y. pestis*. His489 and Asp885 in RcsC are the predicted autophosphorylation and subsequent transfer receipt sites, respectively (*Clarke et al., 2002*; *Latasa et al., 2012*). A D885A mutation in RcsC abolished its function in an $rcsD_{pstb}$ substitute or wild type strain (*Figure 3F–H*), indicating that Asp885 was crucial for phosphate transfer in both conditions. An H489A mutation in RcsC did not alter its function in the wild type strain (*Figure 3F and H*), indicating that His489 is not important for the phosphorylation of RcsD-Hpt. Surprisingly, mutation of His489 resulted in decreased biofilm formation and CR adsorption in the $rcsD_{pstb}$ substitution strain (*Figure 3G and H*), indicating that His489 might function as a phosphate reservoir involved in dephosphorylation of RcsD. RcsC T903A constitutively activates Rcs in *S. enterica* (*García-Calderón et al., 2005*; *Latasa et al., 2012*). The corresponding T913A mutation in RcsC strongly decreased biofilm formation and CR pigmentation in the $RcsD_{pstb}$ substitution strain (*Figure 3F–H*), indicating that an RcsC T913A mutant stimulates Rcs in a similar manner to *S. enterica*. The same mutation in RcsC resulted in enhanced biofilm formation in *Y. pestis*, indicating that an RcsC T913A mutant has an impaired ability to phosphorylate RcsD-Hpt. Taken together, these data indicate that RcsC remains a crucial component of the Rcs phosphorelay system in *Y. pestis*.

RcsB is a phosphoacceptor in the Rcs system and contains a conserved Asp site in its receiver domain which can be phosphorylated by RcsD (*Fredericks et al., 2006*; *Takeda et al., 2001*). Phosphorylated RcsB regulates expression of biofilm-related genes (*hmsT, hmsD, hmsP,* and *hmsHFRS*) in *Y. pestis* (*Fang et al., 2015*; *Guo et al., 2015*; *Sun et al., 2012*). Expression of $rcsD_{pe}$ or $rcsD_{pstb}$ only conferred altered biofilm formation in the presence of RcsB (*Figure 4A*), indicating that $RcsD_{pe}$ and $RcsD_{pstb}$ might differentially modulate the phosphorylation of RcsB. To test this hypothesis, we detected the phosphorylation of RcsB using Phos-tag SDS-PAGE gels and western blotting (*Madec et al., 2014*). As expected, replacement of $rcsD_{pe}$ by $rcsD_{pstb}$ in *Y. pestis* resulted in decreased phosphorylation of RcsB, while mutation of the conserved Asp residue (D56Q) in RcsB abolished phosphorylation (*Figure 4B*, see *Figure 4—source data 1* for details). Consistent with the phosphorylation status of RcsB, the transcription and expression of HmsT were differentially regulated by $RcsD_{pe}$ and $RcsD_{pstb}$ (*Figure 4C and D*, see *Figure 4—source data 2* for details).

## The frameshift mutation in *rcsD* promotes retention of the *pgm* locus during *Y. pestis* flea infection

Loss of function in *rcsA* is a crucial step for *Y. pestis* to establish flea-borne transmission (*Sun et al., 2008*; *Sun et al., 2014*). We therefore speculated that mutation of *rcsD* might also play a role in the

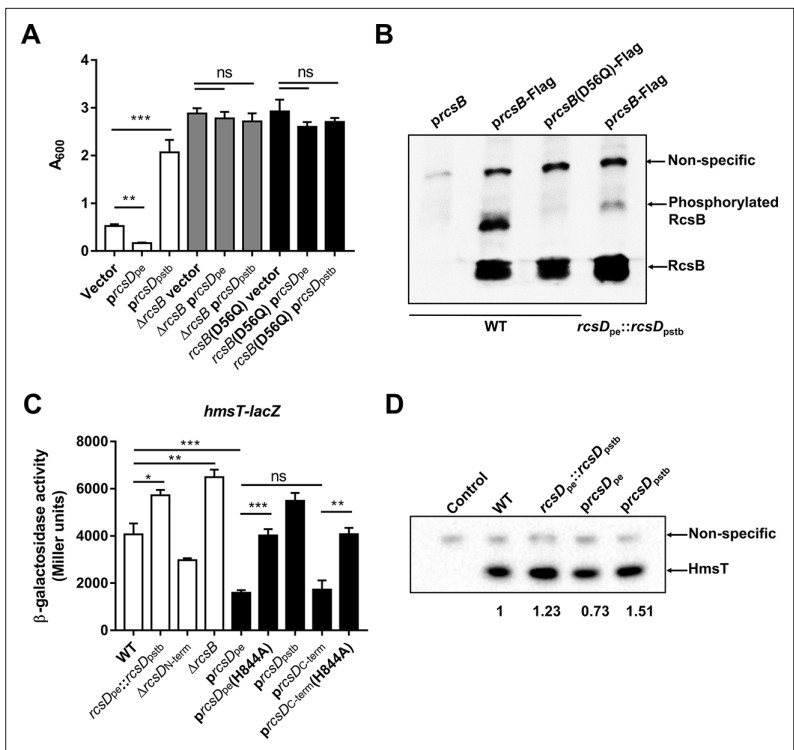

**Figure 4.** The frameshift mutation in *rcsD* increases the phosphorylation of RcsB and represses the expression of HmsT. (**A**) Crystal violet (CV) biofilm assays using a wild type (WT) (white), Δ*rcsB* (grey), or *rcsB* (D56Q) (black) mutant strain, each harbouring pUC19 vectors expressing *rcsD*pe or *rcsD*pstb. (**B**) Phosphorylation analysis of RcsB in the *Y. pestis* KIM6+ strain (WT) harbouring an RcsB expression plasmid (p*rcsB*), a plasmid expressing RcsB fused with a 3xflag tag (p*rcsB*-Flag), or a modified p*rcsB*-Flag expression plasmid in which the conserved phosphorylation site Asp50 was mutated to Gln, and the *Y. pestis* rcsD pstb substitution strain harbouring the p*rcsB*-Flag expression plasmid (see *Figure 4—source data 1* for details). (**C**) Quantification of HmsT expression using a β-galactosidase assay. The *lacZ* reporter gene was fused with the *hmsT* promoter in plasmid pGD926. (**D**) Expression of HmsT was analysed by western blotting using an anti-Flag antibody (see *Figure 4—source data 2* for details). Error bars represent ± SD from three independent experiments with three replicates. Statistical analysis was performed using one-way analysis of variance (ANOVA) with Dunnett's multiple comparisons post-test. ns, not significant; *p<0.05, **p<0.01, ***p<0.001.

The online version of this article includes the following source data for figure 4:

**Source data 1.** Raw source data for *Figure 4B*.

**Source data 2.** Raw source data for *Figure 4D*.

adaptation of *Y. pestis* to the flea. We therefore infected the Oriental rat flea, *Xenopsylla cheopis*, with *Y. pestis* wild type (KIM6+), the *rcsD*pe::*rcsD*pstb and the *rcsD*N-term deletion strain. Bacterial burdens in infected fleas at 0, 7, and 28 day (s) post infection were not significantly different between any strain combinations (*Figure 5A*). In addition, we did not observe significant differences in flea blockage, despite different *in vitro* capacities to form biofilms in these strains (*Figure 5B*). Taken together, these results suggest that the frameshift mutation in *rcsD* does not alter the infection, persistence, and blockage-forming capacity of *Y. pestis* in fleas.

We fortuitously observed that a *Y. pestis* rcsD pstb strain, but not the wild type, displayed a *pgm*-phenotype after 4 weeks of flea infection. The pigmentation phenotype (*pgm*+) is defined by *Y. pestis* absorption of exogenous hemin or CR to form pigmented colonies. A spontaneous deletion of the 102 kb *pgm* locus imposed by the instability of two flanking IS elements results in a *pgm*- phenotype (*Fetherston et al., 1992*; *Tong et al., 2005*). Interestingly, an accumulation in *pgm* locus mutations was previously reported for strains exhibiting enhanced biofilm formation (*Fetherston et al., 1992*; *Kirillina et al., 2004*; *Silva-Rohwer et al., 2021*). Prior to the infectious blood meal, no colonies exhibited a *pgm*- phenotype when the wild type and *rcsD*pstb strains were plated on CR agar plates.

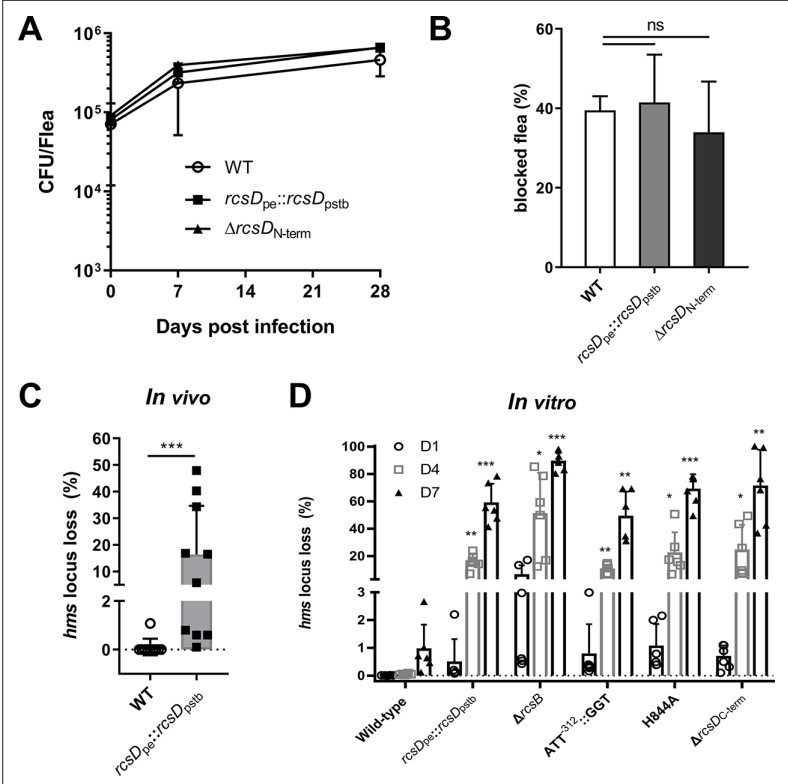

**Figure 5.** The frameshift mutation in *rcsD* stabilizes the *pgm* locus in *Y. pestis*. (**A**) Bacterial burdens in fleas infected with *Y. pestis* wild type (WT), rcsD$_{pe}$::rcsD$_{pstb}$ and *rcsD*$_{N-term}$ strains after 0, 7, and 14 days of infection. (**B**) Cumulative blockage of fleas after 4 weeks of infection with *Y. pestis* WT, rcsD$_{pe}$::rcsD$_{pstb}$ and ΔrcsD$_{N-term}$ strains. Two independent infection experiments are shown. (**C**) Percent of *pgm* locus loss in fleas infected with *Y. pestis* WT and rcsD$_{pe}$::rcsD$_{pstb}$ after 4 weeks of infection. Ten infected fleas were used for this assay. Statistical analysis was performed using a Fisher's exact test. (**D**) Percent of *pgm* locus loss *in vitro* with *Y. pestis* KIM6+ and mutants. Two-way analysis of variance (ANOVA) with Dunnett's multiple comparisons were performed for statistical analysis of mutants with WT strain KIM6+. Error bars represent ± SD from three independent experiments with six replicates. ns, not significant; *$p<0.05$, **$p<0.01$, ***$p<0.001$.

Twenty-eight days post infection, we observed the *pgm-* phenotype in bacteria isolated from all fleas infected with the rcsD$_{pe}$::rcsD$_{pstb}$ strain. The mean percentage of isolates displaying the *pgm-* phenotype was 16.3%, ranging from 0.1% to 47.9%, whereas only one out of ten fleas infected with the wild type strain displayed this phenotype with a frequency of ~1.0% (**Figure 5C**). We further analysed the *pgm* mutation rate of *Y. pestis* in liquid medium. To mimic the environment in the flea, *Y. pestis* were grown in liquid medium for several days, where they remained in stationary phase before reinoculation. Consistent with the flea infection data, *Y. pestis* encoding *rcsD*$_{pstb}$ displayed a significantly higher *pgm-* mutation rate relative to the wild type strain (**Figure 5D**). Strains with deletion of *rcsD-Hpt* or mutated *rcsD*$_{pe}$ also exhibited increased *pgm* mutation rates (**Figure 5D**), indicating that RcsD-Hpt is important for stability of the *pgm* locus. This data agrees with previous reports of strains showing increased *pgm* locus loss concomitant with enhanced biofilm production levels (**Fetherston et al., 1992**; **Kirillina et al., 2004**; **Silva-Rohwer et al., 2021**). PCR analysis confirmed that the *pgm-* phenotype was caused by the spontaneous deletion of the *pgm* locus (data not shown). Taken together, these results suggest that the frameshift mutation in *rcsD* promoted stable maintenance of the *pgm* locus of *Y. pestis* KIM6+ in infected fleas.

## Genome-wide identification of genes regulated by the Rcs phosphorelay system in *Y. pestis*

Rcs has been reported to modulate virulence in other bacteria (**Li et al., 2015**; **Wall et al., 2018**; **Wang et al., 2012**). KIM6+ is an avirulent derivative of the fully virulent KIM strain, which was cured

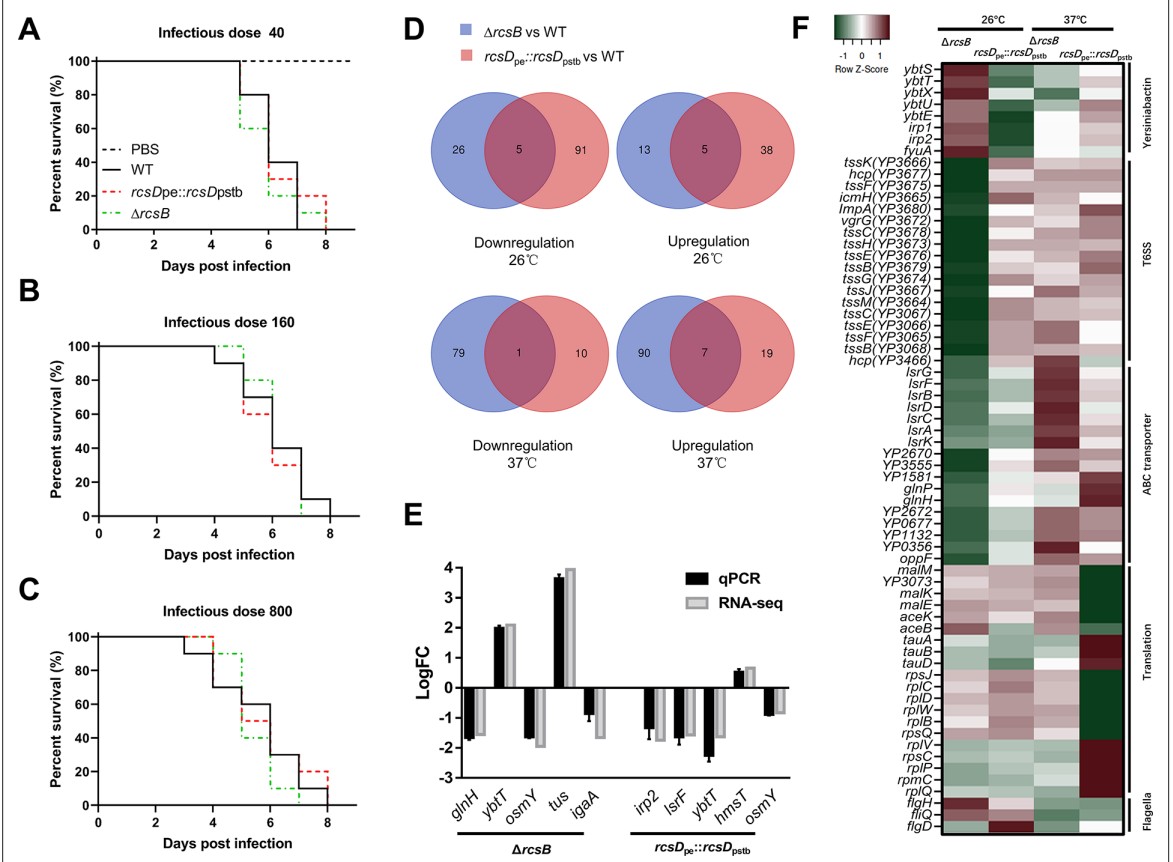

**Figure 6.** Genome-wide identification of genes regulated by the Rcs phosphorelay system in *Y. pestis*. (A–C) Survival of C57BL/6 mice infected with *Y. pestis* Microtus strain 201 and its derivatives using an infectious dose of 40, 160, and 800 colony-forming units (CFU). (D) Venn diagram of upregulated and downregulated genes in *Y. pestis* strains growing at temperatures (26°C or 37°C). (E) qPCR analysis of *hmsT* and differentially expressed genes (DEGs) identified by RNA-seq. The screening threshold for DEGs was defined as |logFC|≥1, and p≤0.05. Error bars indicate SD from at least three samples. (F) Heatmaps showing the differential expression of genes identified through clusters of orthologous group (COG) analysis.

The online version of this article includes the following figure supplement(s) for figure 6:

**Figure supplement 1.** RNA-seq analysis of genes regulated by Rcs system, related to *Figure 6*.

of the pCD1 plasmid. To investigate the role of Rcs on pathogen virulence in a mammalian host, we took advantage of the *Y. pestis* biovar Microtus strain 201 (*Zhang et al., 2014*), a human-avirulent but rodent-virulent strain, isolated from a natural reservoir, the Brandt's vole (*Microtus brandti*) (*Song et al., 2004*). Mutation of *rcsB* or *rcsD* in the *Y. pestis* biovar Microtus strain 201 showed a similar CR absorption and *in vitro* biofilm phenotype as the *Y. pestis* KIM6+ strain (data not shown). To our surprise, deletion of *rcsB*$_{pe}$ or replacement of *rcsD*$_{pe}$ by *rcsD*$_{pstb}$ in this strain did not significantly affect their virulence when mice were subcutaneously infected with different doses of bacteria (*Figure 6A–C*).

Rcs has been reported to modulate the expression of many genes in response to environmental stress. To characterize the genes regulated by the Rcs system in *Y. pestis*, we performed RNA-seq on total RNA isolated from *Y. pestis* biovar Microtus strain 201, *rcsB* deletion and *rcsD* substitution strains cultured at 26°C and 37°C. A total of 139 genes (43 upregulated and 96 downregulated) and 49 genes (18 upregulated and 31 downregulated) were significantly differentially expressed (|logFC|≥1, p≤0.05) in the *rcsB* deletion and *rcsD* substitution strain, respectively, when compared with the wild type strain at 26°C (*Figure 6D, Figure 6—figure supplement 1A–1B*, and *Supplementary file 4*). At mammalian temperature (37°C), 37 genes (26 upregulated and 11 downregulated) and 177 genes (97 upregulated and 80 downregulated) were significantly differentially expressed (*Figure 6D, Figure 6—figure supplement 1C–1D* and *Supplementary file 4*). Several differentially expressed genes (DEGs) identified by RNA-seq were verified by quantitative real-time PCR (qRT-PCR), indicating comparable patterns of expression (*Figure 6E* and *Supplementary file 4*). Substitution of *rcsD*$_{pstb}$ had an opposing

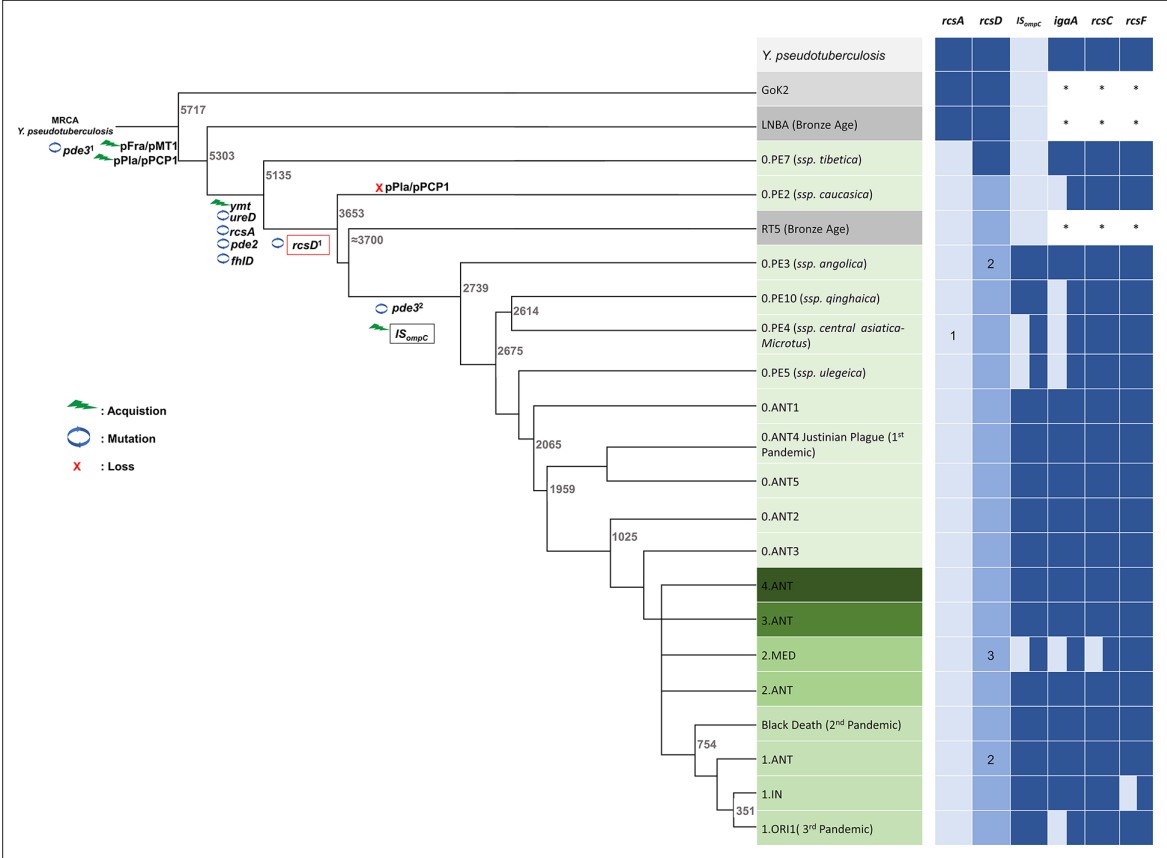

**Figure 7.** Genetic changes in Rcs genes during speciation of *Y. pestis*. Acquisition events are indicated by green lightning symbols, loss of genetic material by a red cross, and mutation by blue circles. All comparisons shown are relative to *Y. pseudotuberculosis*. Increasingly darker shades of grey represent *Y. pseudotuberculosis*, and ancient strains from the Iron Age and the LNBA lineage, respectively. Increasingly darker shades of green represent branches one to four as annotated in the figure. Rcs-related genes and IS*_{ompC}* were considered as intact (dark blue), mutated (light blue), or absent (white). *: not analyzed. 1, present in all *Y. pestis* except branch 0 strain 0.PE4b; 2, *rcsD*_{N-term} and *rcsD*_{C-term} were located in a different genome site in the Nairobi (1.ANT), Angola (0.PE3), and Algeria3 (ORI) strains due to chromosome rearrangement; 3, indels are present in *rcsD*_{N-term} in some strains I-3086 (0.PE4m) and A-1825 (2.MED1). MRCA, the most recent common ancestor. ***Figure 7*** and nomenclature are adapted from Figure 1 of ***Demeure et al., 2019***.

effect on gene expression to deletion of *rcsB* (***Figure 6F*** and ***Supplementary file 4***), indicating that the frameshift mutation in *rcsD* lessens the regulatory function of Rcs. Furthermore, Rcs positively regulated genes such as those encoding the type 6 secretion system, and those related to biosynthesis of yersiniabactin and ABC transporter genes (***Figure 6F***). These observations suggest that Rcs might play an important role in the environmental fitness and virulence of *Y. pestis*, and thus could be required for the flea-mammalian host transmission cycle in the wild.

## A frameshift mutation in *rcsD* is an evolutionary step present in modern *Y. pestis* lineages

To investigate the role of the *rcsD* mutation in the evolution of *Y. pestis*, we analysed the evolutionary changes that occurred during the divergence of *Y. pestis* from *Y. pseudotuberculosis* (***Figure 7*** and ***Supplementary file 5***). Mutation of *rcsD* is present in all *Y. pestis* that harbour five genetic changes (*pde3'*, *ymt*, *rcsA*, *pde2*, and *ureD*) required for flea colonization (***Figure 7***), except for the 0.PE7 branch (***Figure 7***). An IS element in the *ompC* gene is one of two IS elements driving instability of the *pgm* locus (***Figure 7*** and ***Supplementary file 5***; ***Fetherston et al., 1992***; ***Tong et al., 2005***). The IS element in *ompC* was present in most modern *Y. pestis* branches, but not in RT5, an ancient *Y. pestis* strain isolated from a bubonic plague patient during the Bronze Age (***Spyrou et al., 2018***). This indicates that emergence of the *rcsD* mutation is likely not due to loss of the *pgm* locus enabled by IS elements. Although multiple *rcsD* mutations are present across the phylogeny (***Figure 7*** and

*Supplementary file 5*), the HPt encoding region is present in all sequenced *Y. pestis* isolates, indicating an important role of RcsD-Hpt in refining stable blockage-mediated flea-borne transmission of *Y. pestis*.

## Discussion

*Y. pestis* and its ancestor *Y. pseudotuberculosis* have historically been studied as models for pathogen evolution, and have helped to shape our understanding of the evolutionary processes driving niche adaptation, transmission, and pathogenesis (*Wren, 2003*). A recent paleogenomic study has clarified major steps driving evolution of *Y. pestis* (*Rasmussen et al., 2015*; *Spyrou et al., 2018*). An ancestral *Y. pseudotuberculosis*, which has a mutation in the promoter region of *pde3* (*Sun et al., 2014*), acquired two plasmids, pPla and pMT1, in addition to other genetic changes, to evolve into the ancient virulent *Y. pestis* (*Bearden et al., 2009*; *Cui et al., 2013*; *Sodeinde et al., 1992*; *Zimbler et al., 2015*). At this point, *Y. pestis* may still have been prevalent in the environment, where it could be transmitted to humans and animals by the faecal-oral route and occasionally by flea bites through early phase transmission (*Figure 8—figure supplement 1*). Later, other genetic changes, including acquisition of *ymt*, and mutations in *rcsA*, *pde2*, and *ureD*, occurred in the ancient *Y. pestis*, converting the pathogen into full competency for a flea-borne transmission modality (*Chain et al., 2004*; *Cui et al., 2013*; *Hinnebusch, 2005*; *Figure 8—figure supplement 1*).

Compared to its ancestor, which faced multiple changing environments, the establishment of a flea-mammalian host transmission cycle limited environmental exposure of *Y. pestis*. The progenitor lineage of flea-borne *Y. pestis* still required a series of genetic changes to repurpose its environmental signal sensing and transduction systems to adapt to its new lifestyle and niche (*Hinnebusch et al., 2017*; *Yang et al.,*

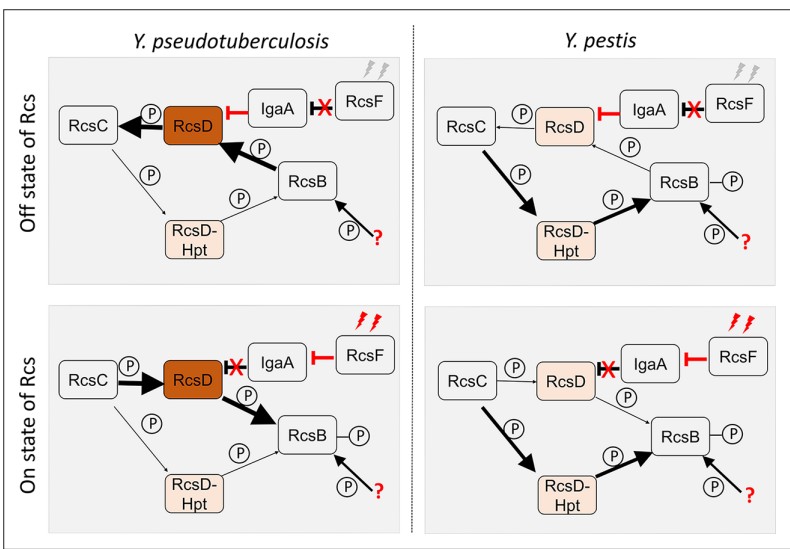

**Figure 8.** Predicted model of Rcs signal transduction and phosphoryl transfer in *Y. pseudotuberculosis* and *Y. pestis*. In *Y. pseudotuberculosis*, full-length RcsD plays a dominant role while the lowly expressed RcsD-Hpt might play a moonlighting function. When Rcs is in 'off' state, RcsF is not activated leading to release from repression of IgaA, which interacts with RcsD. In this situation, RcsB, which might receive the phosphoryl group from another source such acetyl phosphate, is dephosphorylated by intact RcsD, which is then dephosphorylated by RcsC. When Rcs is in an 'on' state, activated RcsF interacts with IgaA, releasing the interaction of RcsD and IgaA. In this situation, RcsC transfers the phosphoryl group to RcsD, and then to RcsB, leading to an activated Rcs system. In *Y. pestis*, RcsD-Hpt might play a dominant role. Rcs system might be only slightly regulated by IgaA and RcsF pathway. In this situation, RcsC transfers the phosphoryl group to RcsD-Hpt, and then to RcsB, leading to an activated Rcs system. The arrows indicate direction of flow of phosphate and weight of the arrows correlates with the flow of amount of phosphate. Weight of the red inhibitor lines correlates with magnitude of inhibition. Phosphate is denoted as small circles with the letter P.

The online version of this article includes the following figure supplement(s) for figure 8:

**Figure supplement 1.** Evolutionary model for how genetic changes in the Rcs system fine-tunes biofilm formation to preserve virulence of flea bite-transmitted *Y. pestis*.

*2023*). Loss of redundant genes and response pathways may have contributed to the fitness of *Y. pestis*. For example, mutation of flagella-related genes in *Y. pestis* occurred in parallel to its colonization of the flea (*Minnich and Rohde, 2007*). Analysis of a broad set of *Yersinia* genomes demonstrated that deletion of one thymine in *rcsD* occurred after the ancient *Y. pestis* acquired the major genetic changes required for flea-borne transmission (*Figures 7 and 8*). The frameshift mutation in *rcsD* might compensate for the fitness cost imposed by loss of *rcsA* function by dampening the subsequent drastic changes in gene expression.

The frameshift present in $rcsD_{pe}$ leads to the expression of two functional proteins: RcsD-Hpt and intact RcsD. Low levels of intact RcsD may be expressed by translational readthrough, while RcsD-Hpt is expressed from a rare AUU start codon. Intact RcsD and RcsD-Hpt might have different functions in *Y. pestis* (*Figure 8* and *Figure 8—figure supplement 1*). Under normal conditions, RcsD dephosphorylates RcsB, while RcsD-Hpt phosphorylates RcsB (*Figure 8*). This subsequently promotes different capacities for biofilm formation and likely multiple other phenotypes. The periplasmic domain of RcsD receives environmental signals sensed by RcsF and IgaA, which in turn regulates the phosphorylation of RcsD by RcsC (*Wall et al., 2018*; *Wall et al., 2020*). RcsD-Hpt lost the ability to respond to environmental signal transduction by RcsF and RcsD but could still receive a phosphate group from RcsC (*Figure 8*). Although RcsD-Hpt and intact RcsD were expressed in *Y. pestis*, RcsD-Hpt appears to play a dominant role in regulation of the Rcs pathway (*Figure 8*). This hypothesis is supported by two observations: (1) expression of $rcsD_{pe}$ conferred a similar phenotype as expression of *rcsD-hpt*, and (2) RcsF and IgaA modulate Rcs signalling in the $rcsD_{pstb}$ substitution strain but not in wild type *Y. pestis*.

Although $rcsD_{pstb}$ has the same RBS and start codon, only a very small amount of RcsD-Hpt relative to full-length RcsD was detected by western analysis, indicating intact RcsD plays a major role in the background of $rcsD_{pstb}$. Sequence analysis indicated that a putative start codon and RBS are present in *rcsD* in many organisms (*Supplementary file 6*). This indicates that RcsD-Hpt may play a moonlighting role in Rcs signalling in *Y. pseudotuberculosis* (*Figure 8*). Indeed, wild type RcsD in *E. coli* produces low levels of a short phosphotransfer protein (*Rogov et al., 2004*; *Wall et al., 2020*). HPt orphan proteins function as phosphate transfer components in multiple phosphorelay systems in numerous prokaryotes and eukaryotes (*Hérivaux et al., 2018*; *Kennedy et al., 2016*; *Mohanan et al., 2017*; *Valentini et al., 2016*), and have evolved from larger phosphotransferase proteins containing multiple domains. The frameshift present in *rcsD* of *Y. pestis* may represent an ongoing evolutionary process generating an orphan HPt protein and consequently a new regulatory pathway.

The *rcsD* frameshift alters the Rcs signalling pathway, which in turn decreases *Y. pestis* biofilm formation. The *rcsD* frameshift in *Y. pestis* does not significantly affect mammalian virulence and flea colonization in our study, but it may promote bacterial fitness during successive flea-mammal host transmission cycles. In agreement with previous findings that spontaneous deletion of the *pgm* locus is increased with enhanced biofilm formation (*Fetherston et al., 1992*; *Podladchikova et al., 2002*; *Silva-Rohwer et al., 2021*), our work shows that increased biofilm formation in an *rcsD* mutant promotes biofilm formation with a subsequent increase in *pgm* loss. The *pgm* locus harbours the *ybt* operon, which is involved in iron acquisition and is required for virulence of *Y. pestis* (*Fetherston et al., 2010*; *Sebbane et al., 2010*). Indeed, *Y. pestis* strains lacking the *pgm* locus are avirulent and have been used as live plague vaccines in some countries (*Bearden and Perry, 1999*; *Podladchikova et al., 2002*). It is notable that prolonged exposure to biofilm-stimulating conditions in the flea gut and during *in vitro* growth resulted in exacerbated loss of the *pgm* locus (*Figure 5C and D*). This may explain why no effect on virulence was noted for the $rcsD_{pe}::rcsD_{pstb}$ strain in our mouse infection studies which utilized strains cultured at 37°C overnight. Enhanced biofilm production through the *rcsA* mutation in the absence of accompanying *rcsD* mutation in ancient *Y. pestis* strains likely caused a high rate of *pgm* locus loss during flea infection thus conferring vaccine-like protection when transferred to the mammalian host. Eventually this would break the transmission cycle between flea and mammalian host. In modern lineages the *rcsD* mutation therefore serves to dampen biofilm production without obvious compromise to flea blockage and infection rates. Thus, stable maintenance of the *pgm* locus required to amplify infection in the mammalian host and perpetuate the flea-mammal transmission cycle, and intensity of plague outbreaks, is promoted (*Figure 7*). Finally, the frameshift mutation of *rcsD* might represent an important step in the emergence of extant ubiquitous lineages of *Y. pestis*.

# Materials and methods

## Bacterial strains and plasmids

This study utilized *Y. pestis* strain KIM6+, which is derived from the sequenced strain KIM strain (***Deng et al., 2002***), but is cured of the pCD1/pYV plasmid required for mammalian virulence, and is competent for flea blockage (***Hinnebusch et al., 1996***) and biofilm formation (***Darby et al., 2002***; ***Sun et al., 2008***). Studies in mice utilized the biovar Microtus strain 201 which was avirulent to human but fully virulent to mice (***Zhang et al., 2014***).

Deletion of *rcsC, rcsF, and igaA* was achieved using a one-step method to integrate PCR products into the chromosome with pKD46, as previously described (***Datsenko and Wanner, 2000***). Double mutants were made by the sequential application of pKD46-mediated deletion. CRISPR-Cas12a-assisted recombineering were used to introduce point mutations, deletions, insertions, and gene replacements in this study (***Yan et al., 2017***). All strains were verified by PCR, DNA sequencing, and plasmid complementation.

For construction of the *hmsT::lacZ* reporter, 350 bp of *hmsT* upstream sequence, together with the first seven codons of the *ORF*, were amplified by PCR using KIM6+ chromosomal DNA as the template. The DNA fragments were digested with HindIII and BamHI restriction enzymes and cloned into pGD926 (***Ditta et al., 1985***; ***Vieille and Elmerich, 1990***), generating plasmids pYC593 and pYC287. Plasmids expressing *rcsC* (D885A), *rcsC* (T913A), *rcsD* (H844A), *rcsD::rcsD-3xflag,* and *rcsD-3xflag::rcsD-3xflag-his6* were generated by overlapping PCR as described previously (***Li et al., 2015***).

For inducible *rcsD* expression, the gene was cloned downstream of the arabinose-inducible promoter of plasmid pBAD/Myc-His (Invitrogen). The plasmid used for determining readthrough was generated by cloning a partial sequence of *rcsD* (159 bp for $rcsD_{pstb}$ and 158 bp for $rcsD_{pe}$) containing the 8T (frameshifted region) into pMal-*lacZ*, generating plasmids pMal-$rcsD_{pstb}$-*lacZ* and pMal-$rcsD_{pe}$-*lacZ*. An additional stop codon was introduced into the 158 bp $rcsD_{pe}$ sequence of pMal-$rcsD_{pe}$-*lacZ*, generating plasmid pMal-$rcsD_{pe}$-stop-*lacZ*.

All strains and plasmids used in this study are shown in *Supplementary file 1* and oligonucleotides used in this study are shown in *Supplementary file 3*.

## *In vitro* biofilms

Microtiter plate biofilm assays were performed as previously described (***Sun et al., 2012***). Briefly, bacteria were cultured overnight in LB broth supplemented with 4 mmol $CaCl_2$ and 4 mmol $MgCl_2$. Cultures were subsequently diluted into 96-well plates and incubated with shaking for 24 hr at 26°C. The wells were washed, and the adherent biofilm was stained with crystal violet, solubilized with 80% ethanol and 20% acetone, and measured by $A_{600}$. Results are from three independent experiments with three technical replicates per experiment.

## β-Galactosidase assays

β-Galactosidase activities were measured as previously described (***Guo et al., 2015***; ***Sun et al., 2012***). Briefly, overnight cultures of *Y. pestis* harbouring *lacZ* reporters were diluted to an $OD_{600}$ of 0.05 and grown in LB broth at room temperature to an $OD_{600}$ of 1.5. The active β-galactosidase, encoded by the *lacZ* reporter gene in *Y. pestis* strains, can cleave o-nitrophenyl-β-D-galactopyranoside (ONPG) substrate to a bright yellow product. The cells were lysed and ONPG solution were added. After incubation at 37°C, the reaction was stopped by adding 1 M $Na_2CO_3$, then absorbance was measured at 420 nm. Results were normalized against cell density and incubation time, and shown in Miller units (***Miller, 1972***). At least two independent experiments with technical triplicates were performed.

## Quantitative real-time PCR

qRT-PCR was carried out as previously described (***Sun et al., 2011***). Briefly, cells were first grown in LB broth overnight before diluting to an $OD_{600}$ of 0.05 in LB and incubating at room temperature to an $OD_{600}$ of 0.8. Total RNA was isolated using the Rneasy Mini Kit (Qiagen). Residual DNA was removed by treatment with rDNase I (Ambion) and confirmed by PCR. cDNA was synthesized from the RNA and used for quantitative PCR on an Applied Biosystems unit (Quant Studio 5). The quantity of mRNA was normalized relative to the reference gene *16sRNA* (YP_r1). The relative mRNA expression levels in each strain were normalized to the wild type samples. Primers and probe sets used in this study are

listed in *Supplementary file 3*. Results from three independent experiments performed in technical triplicate were analyzed by one-way analysis of variance (ANOVA) with Bonferroni's test.

## Western blotting

Western blotting was performed as previously described (*Ren et al., 2017*). For detection of enriched RcsD$_{pe}$, *Y. pestis* strains expressing *rcsD* fused with 3xFlag-His6 were grown at 26°C to stationary phase. Cells were harvested by centrifugation and disrupted by sonication. The protein was enriched by Ni-nitrilotriacetic acid His resin for western blot analysis. For detection of enriched RcsD and RcsD-Hpt in *Figure 2—figure supplement 1E and F*, *Y. pestis* strains expressing *rcsD* fused with 3xFlag were grown at 26°C to stationary phase. Cells were harvested by centrifugation and disrupted by sonication. The protein was enriched by flag resin for western blot analysis. For detection of HmsT, *Y. pestis* strains harbouring a plasmid expressing HmsT with 3xFlag were grown at 26°C to stationary phase. Cells were harvested by centrifugation and disrupted by sonication. Approximately 20 ng proteins were loaded for detection of HmsT. The proteins were separated on 10% SDS-PAGE gels transferred to PVDF membranes (Millipore), analyzed by immunoblotting with an anti-Flag antibody produced by Invitrogen (Catalog number: MA1-91878-HRP; RRID: AB_2537626), and detected with ECL Western Detection Reagents (Bio-Rad). Resulting bands were quantitated by densitometry using NIH ImageJ (*Gallo-Oller et al., 2018*).

## Phos-tag SDS-PAGE

For detection of protein phosphorylation, acrylamide gel was mixed with 25 µM Phos-tag acrylamide (AAL-107, Wako) and 25 µM MnCl$_2$ (*Madec et al., 2014*). Cells grown to an OD$_{600}$ of 0.8 were centrifuged and pellets were resuspended in PBS containing Protease Inhibitor Cocktail (Roche) and then lysed with a sonicator. Samples were quickly loaded onto gels containing 25 µM Phos-tag acrylamide and 25 µM MnCl$_2$ and run. After a 10 min wash with WB transfer buffer supplied with 1 mM EDTA, followed by a 10 min wash with transfer buffer without EDTA. Proteins were transferred to a PVDF membrane and blotted using an anti-Flag antibody produced by Invitrogen (Catalog number: MA1-91878-HRP; RRID: AB_2537626).

## RNA-seq

Total RNA was extracted using the RNeasy Kit (Qiagen). RNA-seq and expression quantification were performed by Genewiz. Gene expression levels were further normalized using the fragments per kilobase of transcript per million mapped reads method to eliminate the influence of different gene lengths and sequencing depth (*Wang et al., 2009*). The edgeR package was used to identify DEGs across samples with fold changes ≥2 and a false discovery rate-adjusted p (p-value)≤0.05 (*Anders and Huber, 2010*; *Robinson et al., 2010*). DEGs were then subjected to an enrichment analysis of GO function and KEGG pathways (*Harris et al., 2004*; *Kanehisa and Goto, 2000*).

## Murine infection

Animals were handled in strict accordance with the Guidelines for the Welfare and Ethics of Laboratory Animals of China and all the animal experiments were approved by the Institutional Animal Care Committee of Military Medical Sciences. Bacterial cultures at 37°C were washed twice with PBS (pH 7.2) and then subjected to serial 10-fold dilutions with PBS. Dilutions were plated onto brain heart infusion (BHI) agar plates to calculate the numbers of colony-forming units (CFU). For each strain, different doses of bacterial suspension were inoculated subcutaneously at the inguinal region of 10 female BALB/c mice (aged 6–8 weeks), which were obtained from Charles River Laboratories (Beijing, China). Survival was monitored at regular intervals, and a survival curve was generated with GraphPad Prism 5.0. *p*-Values were determined using the log-rank (Mantel–Cox) test and the Gehan–Breslow–Wilcoxon test; p<0.01 was considered statistically significant.

## Flea blockage

Flea infections and blockage analysis were carried out as previously described (*Silva-Rohwer et al., 2021*; *Sun et al., 2014*). *Y. pestis* strains were grown overnight in 3 mL HIB at 26°C with shaking, then diluted into 100 mL HIB to cultivate at 37°C without shaking. The following day harvested cells were suspended in sterile PBS, and optical density at absorbance of 600 nm was determined. A

commercial preparation of heparinized mouse blood (BioIVT, New York) was inoculated with *Y. pestis* to a final concentration of CFU/mL of $\sim 5 \times 10^8$ to $1 \times 10^9$. *X. cheopis* fleas were allowed to feed on the infected blood through a mouse skin membrane. Studies with mice were performed in strict accordance with the U.S. National Institutes of Health (NIH) Guide for the Care and Use of Laboratory Animals (National Research Council *Committee for the Update of the Guide for the Care and Use of Laboratory Animals et al., 2011*) and as approved by the Washington State University Institutional Animal Care and Use Committee.

Mice: A CD-1 mouse breeding colony originally sourced from Envigo (https://www.envigo.com/model/hsd-icr-cd-1) is maintained at WSU. Males and females are used. Neonates between the ages of 2–6 days are used for feeding fleas for breeding and maintenance of infected fleas.

Fleas: *X. cheopis* fleas are maintained at WSU since 2010 in Dr. Vadyvaloo's lab. These fleas were originally sourced from Dr Joseph Hinnebusch's lab at the NIH. Males and females were used for experiments.

## Analysis of pigmentation

For *in vivo* pigmentation phenotype detection, 10 fleas infected with *Y. pestis* wild type or the *rcsD*-pe::*rcsD*pstb strain were collected at the end of the infection period (T=28 days). Fleas were individually triturated in sterile PBS, and the fractions were serially diluted and cultivated on a CR plate. Two days later, CFU enumeration relevant to the pigmentation phenotype was performed. For *in vitro* pigmentation assays, fresh *Y. pestis* KIM6+ strains were inoculated into BHI medium and grown at 26°C for 24 hr. Cultures were diluted 1:100 into fresh BHI medium every 3 days. The cultures at 0, 1, 4, and 7 days post inoculation were plated on CR plates for analysis. The loss of the *pgm* locus was confirmed by PCR using two sets of primers described in *Supplementary file 3*. One set of primers targets the *hmsS* gene, which can obtain PCR products from the wild type strain but not the *pgm* locus deletion mutant. Another set of primers target the upstream and downstream regions of the *pgm* locus, and can be used to obtain PCR products from the *pgm* locus deletion mutant but not the wild type strain.

## Quantification and statistical analysis

Figure legends detail the quantification and statistical analyses methods. We conducted the statistical analyses by GraphPad Prism.

## Acknowledgements

We thank Dr. Xiaoyun Pang at Institute of Biophysics, Chinese Academy of Sciences (Beijing, China) for help to predict the structure of RcsD-Hpt by AlphaFold2.

## Additional information

### Funding

| Funder | Grant reference number | Author |
|---|---|---|
| National Key R&D Program of China | 2022YFC2303200 | Xiao-Peng Guo |
| National Natural Science Foundation of China | 31700072 | Xiao-Peng Guo |
| National Natural Science Foundation of China | 31670139 | Yi-Cheng Sun |
| National Natural Science Foundation of China | 31800120 | Hai-Qin Yan |
| Chinese Academy of Medical Sciences | Non-profit Central Research Institute Fund 2019HY310001 | Yi-Cheng Sun |

| Funder | Grant reference number | Author |
| --- | --- | --- |
| Chinese Academy of Medical Sciences | Innovation Fund for Medical Sciences (CIFMS) 2021-I2M-1-043 | Yi-Cheng Sun |
| Fundamental Research Funds for the Central Universities | 3332021092 | Yi-Cheng Sun |
| National Institutes of Health | Research Project Grant R01AI117016-01A1 | Viveka Vadyvaloo |

The funders had no role in study design, data collection and interpretation, or the decision to submit the work for publication.

## Author contributions

Xiao-Peng Guo, Hai-Qin Yan, Conceptualization, Data curation, Formal analysis, Funding acquisition, Validation, Investigation, Visualization, Writing – original draft, Writing – review and editing; Wenhui Yang, Conceptualization, Data curation, Formal analysis, Validation, Investigation, Visualization; Zhe Yin, Formal analysis; Viveka Vadyvaloo, Conceptualization, Resources, Supervision, Funding acquisition, Writing – review and editing; Dongsheng Zhou, Conceptualization, Resources, Supervision, Project administration, Writing – review and editing; Yi-Cheng Sun, Conceptualization, Resources, Supervision, Funding acquisition, Writing – original draft, Project administration, Writing – review and editing

## Author ORCIDs

Xiao-Peng Guo http://orcid.org/0000-0001-5745-2866
Hai-Qin Yan http://orcid.org/0000-0003-1063-3840
Viveka Vadyvaloo http://orcid.org/0000-0003-4842-0525
Yi-Cheng Sun http://orcid.org/0000-0002-5790-7071

## Ethics

The animal study of flea blockage, related to Figure 5A, 5B and 5C, was performed in strict accordance with the U.S. National Institutes of Health (NIH) Guide for the Care and Use of Laboratory Animals (National Research Council Committee for the Update of the Guide for the Care and Use of Laboratory, 2011) and as approved by the Washington State University Institutional Animal Care and Use Committee, under the Animal Subject Approval Form (ASAF) 6641 and 6396. The animal study of murine infection, related to Figure 6A, 6B and 6C, was performed in strict accordance to the Guidelines for the Welfare and Ethics of Laboratory Animals of China and all the animal experiments were approved by the Institutional Animal Care and Use Committee (IACUC) of Academy of Military Medical Sciences (AMMS), ethical approval number IACUC-DWZX-2021-057.

## Decision letter and Author response

Decision letter https://doi.org/10.7554/eLife.83946.sa1
Author response https://doi.org/10.7554/eLife.83946.sa2

# Additional files

## Supplementary files

- Supplementary file 1. Strains used in this study.
- Supplementary file 2. Plasmids used in this study.
- Supplementary file 3. Primers and oligos used in this study.
- Supplementary file 4. Differential expression of *Y. pestis* genes in *Y. pestis* mutant strains growing at temperatures (26°C or 37°C).
- Supplementary file 5. Mutations in *Y. pestis* during evolution involved in this study.
- Supplementary file 6. Putative SD and start codon sites in *rcsD* in Enterobacteriaceae.
- MDAR checklist

## Data availability

All data is available within the paper, its Supporting Information files, and the NCBI GenBank. RNA-seq sequencing data can be accessed in NCBI GenBank using BioProject ID: PRJNA876755. Source data files have been provided for Figures 2D, 4B, 4D, Figure 2—figure supplement 1E and F.

The following dataset was generated:

| Author(s) | Year | Dataset title | Dataset URL | Database and Identifier |
|---|---|---|---|---|
| Guo X-P | 2022 | Differential expression of *Y. pestis* genes in *Y. pestis* mutant strains growing at 26 degrees C and 37 degrees C | https://www.ncbi.nlm.nih.gov/bioproject/PRJNA876755 | NCBI BioProject, PRJNA876755 |

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
