## [Editor Report]

This is a valuable study that provides convincing evidence for fine-tuning of a signal transduction pathway in the emergence of the bacterial pathogen *Yersinia pestis*. The work advances our understanding of bacterial signal transduction, and will be of interest to those studying the evolution of *Y. pestis* and its adaptation to flea-borne transmission.

---

## [Decision Letter]

**Decision letter after peer review:**

Thank you for submitting your article "A frameshift in Yersinia pestis rcsD leads to expression of a small HPt variant that alters canonical Rcs signalling to preserve flea-mammal plague transmission cycles" for consideration by *eLife*. Your article has been reviewed by 3 peer reviewers, including Joseph Wade as Reviewing Editor and Reviewer #1, and the evaluation has been overseen by Dominique Soldati-Favre as the Senior Editor. The following individual involved in review of your submission has agreed to reveal their identity: David Erickson (Reviewer #2).

The reviewers felt that the paper is impactful, the experiments are well done, and most of the conclusions are supported by the data. We have compiled a list of essential revisions that require only changes to the text, and one non-essential revision that would involve an additional experiment, but one that we think is easy to do.

Essential revisions:

1. Conclusions about the specific protein products produced in Y. pestis and Y. pseudotuberculosis are somewhat overstated and should be softened. Data in Figure 2d support the conclusion that Y. pestis makes RcsD-Hpt, but it is less clear whether full-length RcsD is made. We also recommend that the authors mention the possibility that Y. pseudotuberculosis also makes RcsD-Hpt, in the absence of data to test this.

2. The discussion of the Rcs phosphorelay in Y. pestis and Y. pseudotuberculosis is confusing, and some conclusions are overstated. The authors should present a clear model for how phosphate is flowing in the various situations and be clear what is speculation and what is supported by experimental evidence.

Non-essential revision:

We suggest that the authors specifically test the role of RcsD-Hpt in maintenance of the pgm locus. The current data suggest a role for RcsD-Hpt, but it would be interesting (and hopefully easy) to test this directly.

*Reviewer #1 (Recommendations for the authors):*

1. I recommend including a summary figure in the main part of the paper that presents a model involving all the key players.

2. The RNA-seq data do not add much value to the paper and distract from the main story.

3. It would be informative (and straightforward) to test whether deletion of the Hpt domain protein leads to increased loss of the pgm locus.

4. The western blot looking for expression of the Hpt domain of RcsD could be made more convincing by adding some controls, e.g., start codon mutant, strain expressing Yptb rcsD.

*Reviewer #2 (Recommendations for the authors):*

Lines 232-236: My understanding is that in other Enterobacteriaciae, RcsD phosphorylates RcsB, thus switching "on" Rcs signaling. The level of RcsB phosphorylation depends on RcsC either phosphorylating or dephosphorylating RcsD. It isn't clear to me how the data in Figure 3 supports the claim that RcsDpstb is dephosphorylating RcsB. This could be made more obvious.

Line 270: RcsC T913A mutation enhances biofilm (ie. more phosphorylated RcsB). Should this read "impaired ability to dephosphorylate RcsD-Hpt"?

Lines 274-276: Phosphorylation of RcsB is lower when the full-length RcsDPSTB is present compared to Hpt-RcsDYP. According to previously published work, in Y. pestis (which doesn't have RcsA), phosphorylated RcsB represses hmsT transcription, thereby reducing biofilm (Sun et al. 2012) but activates hmsD, thereby increasing biofilm (Guo et al. 2015). Here, the effect of increasing RcsB phosphorylation (Hpt-RcsDYP) is reduced hmsT expression and less biofilm. I cannot find any evidence for RcsB homodimers regulating hmsHFRS directly. This should be clarified.

Figure 5: The rationale for testing the rcsD N-terminal deletion but not the C-terminal Hpt deletion in fleas is not clear.

Line 381: The available evidence from this work suggests that Rcs signaling mutants are not affected in their ability to infect or block fleas or to cause lethal infection in mice. Where else could it be important? It seems possible that the effect of Rcs signaling to induce type 6 secretion might be relevant in the context of infecting wild fleas that have more intact microbiota than the lab-reared fleas. The meaning of this sentence is not clear at present.

Line 424: What mutations have occurred in ymt?

Line 431: The reference cited (Laayouni 2014) does not support the claim that genetic changes altered signal transduction systems that helped Y. pestis adapt to new niches.

Line 471-475: What evidence in Figure 5d supports the claim that biofilm formation increases loss of pgm more than planktonic growth? I cannot find the data in the Silva-Rhower 2021 paper that links loss of RcsD with biofilm production and pgm loss.

*Reviewer #3 (Recommendations for the authors):*

1. A possible interesting question, an extension of this work has to do with whether the Y. pestis pathway is regulated at all. The Rcs pathway in other organisms mainly responds to membrane stress via RcsF. If the RcsDpe is decoupled from IgaA and RcsF, does this pathway respond in any way to any kind of cell envelope stress? Are there any conditions under which the phosphorylation of RcsB should be shut down (by signalling to RcsC, possibly) or further activated?

2. Figures cannot be read except when much enlarged. The Crystal violet plates are hard to interpret and could be moved out of the main figures without losing anything.

3. The pgm phenotype needs to be defined when introduced, and it would be useful to immediately introduce the information that is already known about the nearby IS and that this happens. It suggests that high biofilm cells are selected against in these assays.

4. Overall, the major points in this paper could be more clearly made with a rearrangement to clarify what the Rcs null phenotype is early and a clearer analysis of how the authors think the Rcs system is working in Yersinia pestis.

---

## [Author Response]

Essential revisions:1. Conclusions about the specific protein products produced in Y. pestis and Y. pseudotuberculosis are somewhat overstated and should be softened. Data in Figure 2d support the conclusion that Y. pestis makes RcsD-Hpt, but it is less clear whether full-length RcsD is made. We also recommend that the authors mention the possibility that Y. pseudotuberculosis also makes RcsD-Hpt, in the absence of data to test this.

We have provided more evidence to show that full-length RcsD is made by RcsD_pe_ (Line 166-168 and Figure 2—figure supplement 1G). In addition, we also provided evidence that RcsD-Hpt might be made by *rcsD_pstb_* (Line 166-168 and Figure 2—figure supplement 1F).

2. The discussion of the Rcs phosphorelay in Y. pestis and Y. pseudotuberculosis is confusing, and some conclusions are overstated. The authors should present a clear model for how phosphate is flowing in the various situations and be clear what is speculation and what is supported by experimental evidence.

Thanks for your suggestion. We have provided a model to show how phosphate is flowing in the various situations (Figure 8—figure supplement 1) and a revised evolutionary model to show how Rcs affects evolution of *Y. pestis* in the main manuscript (Figure 8).

Non-essential revision:We suggest that the authors specifically test the role of RcsD-Hpt in maintenance of the pgm locus. The current data suggest a role for RcsD-Hpt, but it would be interesting (and hopefully easy) to test this directly.

Thanks for your suggestion. We have provided the data to show that RcsD-Hpt is required to maintenance of the *pgm* locus (Line 295-296 and Figure 5D).

Reviewer #1 (Recommendations for the authors):1. I recommend including a summary figure in the main part of the paper that presents a model involving all the key players.

Thanks for your suggestion. We have provided a detailed revised model in the main part (Figure 8) which contained all key players.

2. The RNA-seq data do not add much value to the paper and distract from the main story.

We agree that the RNA-seq data do not add much value to the main points of the paper. However, we believe the RNA-seq data provides useful information about Rcs regulated genes and differential gene expression due to the *rcsD* mutation. Thus, we would prefer to keep it.

3. It would be informative (and straightforward) to test whether deletion of the Hpt domain protein leads to increased loss of the pgm locus.

Thanks for your advice. We have done the experiment as suggested and found deletion of Hpt leads to increased loss of the *pgm* locus (Line 295-296 and Figure 5D).

4. The western blot looking for expression of the Hpt domain of RcsD could be made more convincing by adding some controls, e.g., start codon mutant, strain expressing Yptb rcsD.

Thanks for your suggestion. We have done the experiment as suggested and the RcsD-Hpt is only expressed by *rcsD*_pe_ but not the *rcsD*_pe_ with the start codon mutation (Figure 2—figure supplement 1F). We also show that Hpt domain is expressed in enriched fractions of RcsD_pstb_ (Figure 2—figure supplement 1F).

Reviewer #2 (Recommendations for the authors):Lines 232-236: My understanding is that in other Enterobacteriaciae, RcsD phosphorylates RcsB, thus switching "on" Rcs signaling. The level of RcsB phosphorylation depends on RcsC either phosphorylating or dephosphorylating RcsD. It isn't clear to me how the data in Figure 3 supports the claim that RcsDpstb is dephosphorylating RcsB. This could be made more obvious.

RcsB can be phosphorylated and dephosphorylated by RcsD (Wall EA, *et al.*, 2020, Plos Genetics, https://doi.org/10.1371/journal.pgen.1008610 and Takeda S., et al., 2001, Mol. Microbiol., https://doi: 10.1046/j.1365-2958.2001.02393.x). Figure 3 provides evidence for how the frameshifted *rcsD* results in an altered Rcs signalling pathway in *Y. pestis*. Figure 1 shows that expression of RcsD_pstb_ in wildtype *Y. pestis* or the RcsD_N-term_ deletion mutant resulted in increased biofilm formation, which indicates that RcsB is dephosphorylated and switched off. In addition, we have provided new data to show that expression of RcsD-H844A mutant caused significantly less biofilm than expression of RcsD, indicating H844 is important for RcsD to dephosphorylate RcsB. Please also see response to Reviewer 3 comment #2.

Line 270: RcsC T913A mutation enhances biofilm (ie. more phosphorylated RcsB). Should this read "impaired ability to dephosphorylate RcsD-Hpt"?

RcsC T913A mutation enhances biofilm which indicates that there is less phosphorylated RcsB. Thus, it should be “impaired ability to phosphorylate RcsD-Hpt”.

Lines 274-276: Phosphorylation of RcsB is lower when the full-length RcsDPSTB is present compared to Hpt-RcsDYP. According to previously published work, in Y. pestis (which doesn't have RcsA), phosphorylated RcsB represses hmsT transcription, thereby reducing biofilm (Sun et al. 2012) but activates hmsD, thereby increasing biofilm (Guo et al. 2015). Here, the effect of increasing RcsB phosphorylation (Hpt-RcsDYP) is reduced hmsT expression and less biofilm. I cannot find any evidence for RcsB homodimers regulating hmsHFRS directly. This should be clarified.

*hmsHFRS* has been suggested to be directed regulated by RcsB, while *hmsP* is indirectly regulated by RcsB (Fang N, *et al.*, Sci. Rep., 2015, https://doi: 10.1038/srep09566.). We have deleted the word “directly” in the revised manuscript (Line 252).

Figure 5: The rationale for testing the rcsD N-terminal deletion but not the C-terminal Hpt deletion in fleas is not clear.

Based on our previous results, *Y. pestis* mutant (for example, *Y. pestis* with highly expressed *hmsT* or *hmsD*) with increased biofilm formation did not cause increased blockage in the flea (Sun YC *et al.*, 2011, PLoS One, https:// doi: 10.1371/journal.pone.0019267). Thus, we did not test the C-terminal deletion mutant which formed increased biofilm in vitro. Because the N-terminal deletion mutant caused decreased biofilm formation in vitro, we asked whether it could cause a reduced amount of blockage in the flea.

Line 381: The available evidence from this work suggests that Rcs signaling mutants are not affected in their ability to infect or block fleas or to cause lethal infection in mice. Where else could it be important? It seems possible that the effect of Rcs signaling to induce type 6 secretion might be relevant in the context of infecting wild fleas that have more intact microbiota than the lab-reared fleas. The meaning of this sentence is not clear at present.

Mutation of Rcs in *Y. pestis* seems not affect the ability of *Y. pestis* to block fleas or to cause lethal infection in mice in the tested conditions. However, we agree that it might have some function during the natural transmission cycle in wild fleas. We have included such a statement in the revised manuscript (Line 334-337).

Line 424: What mutations have occurred in ymt?

Sorry for the mistake. We refer to the acquisition of *ymt*. We have corrected this mistake.

Line 431: The reference cited (Laayouni 2014) does not support the claim that genetic changes altered signal transduction systems that helped Y. pestis adapt to new niches.

We have corrected this oversight by providing an appropriate citation in the revised manuscript (Line 375-376).

Line 471-475: What evidence in Figure 5d supports the claim that biofilm formation increases loss of pgm more than planktonic growth? I cannot find the data in the Silva-Rhower 2021 paper that links loss of RcsD with biofilm production and pgm loss.

Compared to the wild type strain, the *rcsB* mutant and *rcsD*_pstb_ strain formed more biofilm, indicating that increased biofilm formation causes increased loss of the *pgm* locus. We did not compare *pgm* loss between biofilm and planktonic bacterial cultures. Although no numerical data has previously been shown for the positive correlation between greater biofilm production and *pgm* loss, this association has been stated as observations in Silva-Rohwer 2021 and Kirillina *et al.* 2004. We have added more references into the manuscript to strengthen our finding of this phenomena for which we provide empirical evidence.

Reviewer #3 (Recommendations for the authors):1. A possible interesting question, an extension of this work has to do with whether the Y. pestis pathway is regulated at all. The Rcs pathway in other organisms mainly responds to membrane stress via RcsF. If the RcsDpe is decoupled from IgaA and RcsF, does this pathway respond in any way to any kind of cell envelope stress? Are there any conditions under which the phosphorylation of RcsB should be shut down (by signalling to RcsC, possibly) or further activated?

Full length of RcsD could respond to the regular signalling pathway, while RcsD-Hpt has lost the ability to respond to IgaA and RcsF (Figure 8—figure supplement 1). In addition, we agree that RcsC might play a role in regulation of this Rcs pathway. However, this needs to be investigated.

2. Figures cannot be read except when much enlarged. The Crystal violet plates are hard to interpret and could be moved out of the main figures without losing anything.

We have provided the original figures in the revised manuscript and apologize for the challenges of evaluating small figures. We prefer to keep the Crystal violet and CR results as one showed quantitative data and the other one showed observational qualitative phenotypes.

3. The pgm phenotype needs to be defined when introduced, and it would be useful to immediately introduce the information that is already known about the nearby IS and that this happens. It suggests that high biofilm cells are selected against in these assays.

Thanks for your suggestion. We have revised the manuscript as suggested and have introduced the known information about *pgm* earlier (Line 346-350).

4. Overall, the major points in this paper could be more clearly made with a rearrangement to clarify what the Rcs null phenotype is early and a clearer analysis of how the authors think the Rcs system is working in Yersinia pestis.

Thanks for your comment, we have put the data of *rcsB* deletion mutant in Figure 1 in the revised manuscript (Figure 1B). In addition, we have provided a model for how Rcs is working in *Yersinia pestis* and *Yersinia pseudotuberculosis* (Figure 8 and Figure 8—figure supplement 1).